



# Short communication: Inverse correlation between radiation damage and fission-track etching time on monazite

Toru Nakajima[1], Shoma Fukuda[1] Shigeru Sueoka[1], Sota Niki[2], Tetsuo Kawakami[3], Tohru Danhara[4], Takahiro Tagami[3]

[1]Tono Geoscience Center, Japan Atomic Energy Agency, 509-5102, Toki, Japan

[2]Geochemical Research Center, The University of Tokyo, 113-0033, Tokyo, Japan

[3]Department of Geology and Mineralogy, Kyoto University, 606-8502, Kyoto, Japan

[4]Kyoto Fission-Track Co. Ltd, 603-8832, Kyoto, Japan

*Correspondence to*: Toru Nakajima (nakajima.toru@jaea.go.jp)

**Abstract**

In this study, we explored the impacts of radiation damage and chemical composition on the etching time of monazite fission-track (MFT). Despite the potential of MFT as an ultra-low-temperature thermochronology, the comprehensive effects of radiation damage and non-formula elements, especially on the etching rate of MFT, remain unexplored, and established analytical procedures are lacking. We quantified the degree of radiation damage ($\Delta_{FHWM}$) of Cretaceous to Quaternary
monazites distributed in the Japan arc through Raman spectroscopy and chemical composition analyses. Subsequently, MFT etching was performed to examine the correlation between these parameters and the etching time.

Estimation of the degree of radiation damage showed an increase in radiation damage corresponding to the cooling age of each geological unit. For example, Monazites from Quaternary geological units, the Toya ignimbrite (ca. 0.1 Ma) and the Kurobegawa granodiorite (ca. 0.8 Ma), have $\Delta_{FHWM}$ of 0.48 and 0.70 cm⁻¹, respectively. In contrast, the Muro ignimbrite (ca.
15 Ma) has a $\Delta_{FHWM}$ of 4.11 cm⁻¹, while Cretaceous granitoids, including the Kibe granite and the Sagawa granite, yielded 7.42 and 6.40 cm⁻¹, respectively. MFT etching of these samples according to the existing recipe (6M HCl at 90°C for 60−90 minutes) was completed at 1200, 860, 210, 120, and 90 minutes for Toya ignimbrite, Kurobegawa granodiorite, Muro ignimbrite, Sagawa granite, and Kibe granite respectively. These outcomes highlight an inverse relationship between MFT etching time and the degree of radiation damage in monazite, while the correlation between MFT etching time and chemical
composition was unclear. The results affirm earlier considerations that the etching rate of MFT is strongly influenced by radiation damage. Conversely, young samples with lower levels of radiation damage exhibit higher chemical resistance, suggesting that existing etching recipes may not adequately etch MFT.





## 1. Introduction

Monazite (LREE(PO$_4$)) commonly contain U and Th, and is therefore considered a potential target for fission-track (FT) dating. In recent years, the annealing experiment of Cf induced tracks indicate that the closure temperature of monazite fission-track (MFT) system range 25–45 °C in the geological timescale ($10^6$–$10^7$ years: Jones et al., 2021). With ongoing advancements in analytical methodologies and understanding of annealing kinetics of MFT, it is expected to become a powerful tool for approaching the thermal history of ultra-low temperature ranges (<50°C). Moreover, owing to the typically high U and Th

concentration (>1000 ppm) in monazite compared to apatite, MFT dating is anticipated to be applicable to the dating of rocks with post-Quaternary formation and/or cooling ages.

Despite these optimistic prospects, there remain numerous unresolved details in the experimental procedures of MFT at present. For example, MFT etching characteristics are not sufficiently explored. Since the FT (daughter) formed by the spontaneous fission of radionuclides (parent) cannot be observed optically (latent track; Paul and Fitzgerald, 1992), it is

necessary to selectively etch the FTs for measurement for the dating (Fleischer and Price, 1964). Because inconsistency in the degree of etching progression can lead to biases in FT counting and FT length measurements using an optical microscope (e.g., Laslett et al., 1984; Ketcham et al., 2015; Tamer et al., 2019; Tamer and Ketcham, 2020), their protocols have been carefully investigated for robust dating and thermal analysis (Laslett et al., 1982; Green, 1988; Yamada et al., 1993, 1995). On the other hand, variations in FT etching rates are commonly observed within the same sample at inter- and intra-grain level for

conventional FT dating methods (e.g., apatite, zircon, titanite), and factors governing etching speed and subsequent etching protocols based on these factors have been proposed. For apatite, differences in chemical resistance result from its chemical composition, particularly the F-Cl-OH contents (e.g., Burtner et al., 1994), leading to varying FT etching rates. On the other hand, for zircon, there is a well-known relationship where etching time and FT density (∝U concentration) inversely correlate within the same sample (e.g., Gleadow et al., 1976), reflecting the decrease in chemical resistance due to accumulated radiation

damage.

It is widely known that MFT can be etched using HCl. Jones et al. (2019) showed that MFT is etched within a certain duration under controlled temperature (6M HCl at 90°C for 60–90 minutes), and subsequent studies have employed this etching condition (Jones et al., 2021, 2023). Similar to other minerals used in FT dating, previous studies have found inter-grain differences in etching rates for monazite, which is attributed to variation in U and Th concentrations and the resulting radiation

damage accumulation (Fayon, 2011; Jones et al., 2019). In previous studies, only aged monazites like the Antsirabe monazite (ca. 500 Ma) and the Harcourt monazite (ca. 370 Ma) were used in experiments (Weise et al., 2009; Jones et al., 2019). Although some experiments pre-annealed the MFT and radiation damage (Weise et al., 2009; Jones et al., 2019, 2021), the actual effects of radiation damage remain largely unexplored. Particularly, younger (Cenozoic) monazites may lack significant radiation damage and might resist existing MFT etching techniques. Additionally, considering that monazite is a phosphate

with various compositions, similar to apatite, the influence of chemical composition on etching rates should be noted.



In this study, five monazites with various levels of radiation damage were collected in order to investigate mainly the effect of radiation damage on FT etching time. The formation ages of the host rocks range from the Cretaceous to the Quaternary. For these monazites, we determined 1) chemical composition, 2) degree of accumulated radiation damage, and 3) MFT etching time. This paper reports an inversely proportional relationship between MFT etching time and radiation damage and provides some discussion on the MFT etching protocols for young monazites.

## 2. Sample description

### 2.1. The Toya ignimbrite (Toya-5b)

The Toya ignimbrite, a siliciclastic eruptive deposit, is distributed in the western region of Hokkaido, Japan (Fig. 1b). Predominantly comprised of pumiceous beds and lag deposits, it originated from the catastrophic eruption of the Toya caldera at ca. 0.1 Ma (reviewed by Tomiya and Miyagi, 2020). Along the Osaru River, the Toya ignimbrite is subdivided into Units 1–6, which are interpreted to have been deposited by pyroclastic flows during a series of intermittent catastrophic eruptions (Goto et al., 2016). The monazite sample employed in this study was separated from the Unit 5b pumice. Niki et al. (2022) documented a zircon and monazite U-Th disequilibrium age of 113.8 $^{+5.4}_{-5.2}$ ka (95% confidence levels) from the Unit 5b pumice. The outcrop, from which the sample (Toya-5b) was collected, exhibits plateau with no overlying layers other than the Kt-2 tephra which was erupted from the Kuttara volcano in southwestern Hokkaido (ca. 40 ka: Yamagata, 1994). Toya-5b yields euhedral and colorless monazite (Fig. 2a), which includes numerous glassy melt inclusions with vapor. Internal microstructure is not clearly identified in backscattered electron (BSE) images indicating a very homogeneous intra-grain composition (Fig. 2b).

### 2.2. The Kurobegawa granodiorite (KRG19-B04)

The Kurobegawa granodiorite is a dioritic intrusive body exposed in the northern Hida Mountain range, central Honshu Island, Japan (Fig. 1c). Zircon U-Pb ages indicate that this is the youngest plutonic rock presently exposed on the surface of the earth (ca. 0.8 Ma; Ito et al., 2013, 2017, Suzuki et al., 2022). This granodiorite is characterized by numerous mafic magmatic enclaves and is subdivided into three units—lower, middle, and upper—based on the areal proportion of mafic magmatic enclaves in an outcrop (Harayama et al., 2000). Monazite in KRG19-B04 was separated from the biotite-hornblende granite of the lower unit. Monazite in KRG19-B04 shows semi-euhedral or anhedral shape, and is commonly colorless (Fig. 2c). It encloses mineral inclusions of quartz, biotite, and pyrite, primary polyphase mineral inclusions, and secondary fluid inclusions. Secondary domains cutting the primary sector-zoned domain are commonly observed in BES images (Fig. 2d).




## 2.3. The Muro ignimbrite (MR3b)

The Muro ignimbrite is a siliciclastic eruptive rock widely distributed to the northern part of the Kii Peninsula, Honshu Island (Fig. 1d). The zircon and apatite FT ages obtained from the black glassy lapilli tuff at the base of the unit indicate that the Muro ignimbrite was originated by an extensive igneous activity at ca. 15 Ma (Iwano et al., 2007, 2009). The monazite sample employed in this study was separated from the siliceous ignimbrite (MR3b) overlying black glassy lapilli tuff which is cropped out at the same location with MR3 (Iwano et al., 2007). Monazite in MR3b is yellowish and euhedral (Fig. 2e). It includes numerous polyphase inclusions composed of quartz, K-feldspar, and magnetite. It shows no zoning to very weak
oscillatory zoning in BSE images.

## 2.4. The Kibe granite (EY137-21)

The Kibe granite is the porphyritic granite body of the Ryoke Belt exposed in the Yanai area, south-eastern Honshu Island (Fig. 1e). The middle Cretaceous zircon U-Pb dates (98.0 ± 1.0 and 97.6 ± 1.0 Ma (2σ)) have been interpreted as an intrusive age of the pluton (Skrzypek et al., 2016). Jones et al. (2023) reported late Cretaceous to Paleocene zircon and apatite U-Th/He
and apatite FT dates. They also conducted MFT dating and calculated central age of 2.2 ± 0.1 Ma (1σ) using the absolute calibration method (Hasebe et al., 2004) including undetermined parameters (e.g., initial FT length). The monazite sample used in this study were separated from the biotite granite (EY137-21) collected from the riverbed 100 m east of EY137A used in Skrzypek et al. (2016) and Jones et al. (2023). Monazite in EY137-21 is yellow- or orange-colored, semi-euhedral crystal (Fig. 2g). Fine inclusions of quartz, biotite, apatite, and zircon are commonly found. Internal structure of oscillatory zoning and/or sector zoning are commonly observed in BSE images (Fig. 2h). Secondary domains cutting the primary microstructures
are found in some grains.

## 2.5. The Sagawa granite (NR24-21)

The Sagawa granite is also the Cretaceous granitic body of the Ryoke Belt exposed in the northern part of the Kii Peninsula, Honshu Island (Fig. 1d). The certain intrusion age is unknown, although middle Cretaceous biotite Rb-Sr (>105 Ma) age was
reported by Ishizuka et al. (1996). Similar to the Kibe granite, Jones et al. (2023) reported late Cretaceous to Eocene zircon and apatite U-Th/He and apatite FT dates, and MFT central age of 1.7 ± 0.1 Ma (1σ). The monazite sample used in this study were separated from the biotite granite (NR24-21) collected from the same outcrop of NR24B used in Jones et al. (2023). Monazite in NR24-21 is yellow- or orange-colored, semi-euhedral crystal (Fig. 2i) with minor mineral inclusions of quartz, biotite, apatite, and zircon. Oscillatory zoning and/or sector zoning are commonly observed in BSE images (Fig. 2j). Secondary
domains cutting the primary microstructure are found in some grains.



## 3. Method

### 3.1. Estimation of crystal structural disorder and chemical composition

The degree of accumulated radiation damage of monazite was estimated using Raman spectroscopic measurement. The Raman band broadening (FWHM (cm$^{-1}$)) of the $v_l$(PO$_4$) peak is widely applied as an index of the crystal structural disorder (Seydoux et al., 2002; Ruschel et al., 2012). On the other hand, Raman band broadening depends not only on the self-irradiation but also on the incorporation of non-formula elements in monazite. Therefore, the degree of the accumulated radiation damage ($\Delta_{FWHM}$) is estimated as a difference between the measured Raman band broadening (FWHM$_{meas.}$) and the chemical band broadening (FWHM$_{calc.}$):

$$\Delta_{FWHM} = FWHM_{meas.} - FWHM_{calc.} \qquad (1)$$

FWHM$_{calc.}$ is calculated using the calibration by Ruschel et al. (2012):

$$FWHM_{calc.} = 3.95 + 26.66 \times (Th + U + Ca + Pb)[apfu] \qquad (2)$$

To exclude possible effects of the focused electron beam on the accumulated radiation damage, Raman spectroscopic measurements were carried out prior to electron probe micro analyses (Meldrum et al. 1996). Since the measurement points were selected before the BSE image observation, 50 points were selected to avoid inclusions and cracks regardless of the internal structure.

### 3.1.1. Sample Preparation

Mineral separation was conducted at Tono Geoscience Center and Kyoto Fission-Track Co. Ltd. Rock samples were crushed using a stainless-steel mortar. For Toya-5b, white pumices (Goto et al., 2016) were picked out from the pumice bed of the Unit 5b, washed, dried, and crushed. Monazite grains were separated using standard heavy liquid and magnetic separation techniques. They were then mounted on an epoxy resin at room temperature and polished with diamond pastes of 3 μm and 1 μm diameter.

### 3.1.2. Raman spectroscopic measurement

The Raman spectrum of monazite samples was measured using a laser Raman spectrometer (JASCO NRS 3100) at Kyoto University at room temperature (ca. 21°C). Green YAG laser (532 nm) was irradiated at the polished monazite surface through an optical density 1 filter, a 0.05 mm slit, and x100 objective lens. A grating with 1800 grooves/mm was used to disperse the light to be analyzed. The laser power at the sample surface was ~ 3.0 mW. The spectrum spans the wavenumber interval of



200–1400 cm$^{-1}$ and was recorded in 4 acquisitions of 10 s at the spectra resolution of 0.01 cm$^{-1}$. The instrument was calibrated using the Ar lamp emission measured every 10 unknown measurements. The accuracy of wavenumber was better than 0.5 cm$^{-1}$.

Measured Raman spectrums were cropped between 800–1200 cm$^{-1}$ including the $v_1(PO_4)$ band, and fitted by a Lorentzian–Gaussian function for determining the peak position and width. Peak fitting was conducted using the Peakfit v4.12 (SYSTAT) software.

### 3.1.3. Electron probe micro analysis

*In-situ* chemical analyses at the same spot as the Raman spectroscopic measurements were done using a field-emission
electron probe microanalyzer JXA-8530F (JEOL) at Tono Geoscience Center. Spot analyses were operated at an accelerating voltage of 15 kV, beam current of 20 nA, and beam diameter of 5 μm. The background counting times were set at half the respective peak counting time. Natural and synthetic minerals (ASTIMEX REEM25−15 and MINM25−53) were used as standards and ZAF correction was applied. The counting times were set as 30 s for REEs, Ca, P, and Y, 60 s for Th and U, and 15 s for Pb and Si. For the measurement of $UO_2$ (UMβ), a peak interference correction by $ThO_2$ (ThMγ) was applied.

### 3.2. Etching of monazite fission-track

The MFT is etchable with HCl (Shukoljukov and Komarov, 1970; Weise et al., 2009; Jones et al., 2019). In this study, we adopted the etching protocol proposed by Jones et al. (2019): 6M HCl at 90°C for 60–90 minutes, with temperature control facilitated by a water bath. Despite the adherence to the specified conditions, some samples exhibited insufficient etching even after 90 minutes, necessitating additional step-etching.
Previous etching experiments have indicated that the etch pit diameter parallel to the crystallographic c-axes (Dpar) upon etch completion is approximately 0.8 μm (Jones et al., 2022). While the termination conditions for FT dating require more careful consideration, for the sake of convenience, this study adopted 0.8 μm as the termination criterion and employed a stepwise etching procedure to estimate total etching duration time ($t_{MFT}$).

### 4. Results

For each sample in this study, 50 Raman spectra and chemical compositions at the same spot were obtained. Representative values are shown in Table 2 and data for each spot in Tables S1–10. The FWHM$_{meas.}$ and non-formula element composition (Th + U + Ca + Pb) for each spot are shown in Figure 3a, and the violin plot of $\Delta_{FWHM}$ calculated from them is shown in Figure



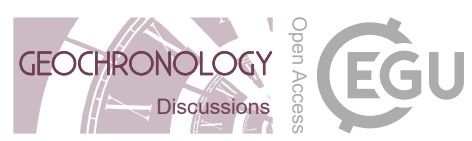

3b. The following sections summarize results of the Raman spectroscopic analysis, electron probe micro analysis, the calculated degree of radiation damage, and MFT etching experiment.

### 4.1. Raman spectroscopic measurement

The $FWHM_{meas.}$ value measured by Raman spectroscopy was significantly smaller for monazites from the younger samples Toya-5b and KRG19-B04, with mean values of 5.19 and 7.07 cm$^{-1}$, respectively. The $FWHM_{meas.}$ value of MR3b (10.54 cm$^{-1}$) was larger than that of the younger samples and tended to be smaller than that of the Cretaceous granites (14.54 and 12.69 cm$^{-1}$ for EY137-21 and NR24-21 respectively). Toya-5b showed very low dispersion, whereas the other four samples showed relatively high dispersion for each measurement spot.

### 4.2. Chemical composition of natural monazite sample

All five monazites were classified as Ce-monazite and showed no significant dispersion in LREE (La-Ce-Nd-Pr) ratios. Monazite in Toya-5b tends to have a very homogeneous inter- and intra-grain composition and is poor in non-formula elements (Fig. 3a). While monazite in MR3b was very homogeneous within each grain, large inter-grain chemical compositional variation was observed. As seen in the BSE images, KRG19-B04, EY137-21, and NR24-21 are diverse in inter- and intra-grain composition. Regarding Th, U, Ca, and Pb concentrations, although variations exist from point to point, these elements tend to be enriched on average in the sequence Toya-5b < NR24-21 < KRG19-B04 < MR3b < EY137-21.

### 4.3. The degree of radiation damage

The degree of radiation damage ($\Delta_{FWHM}$) estimated for monazites in each sample based on equation (1) is shown in Table 2 and Figure 3b. The mean $\Delta_{FWHM}$ values are as follows, arranged in ascending order: 0.48 cm$^{-1}$ for Toya-5b, 0.70 cm$^{-1}$ for KRG19-B04, 4.11 cm$^{-1}$ for MR3b, 7.42 cm$^{-1}$ for EY137-21, and 6.40 cm$^{-1}$ for NR24-21. This relationship is roughly consistent with that of sample ages (Table 1). A degree of dispersion within the sample was observed for four samples (KRG19-B04, MR3b, EY137-21, and NR24-21) which showed variations in chemical composition. These values suggest a progressive increase in structural disorder among the samples due to the accumulation of radiation damage. Although some spots had negative $\Delta_{FWHM}$ values for monazites in KRG19-B04, we considered this to be due to error propagation in calibration of chemical broadening by Ruschel et al. (2012) (equation (2)) and did not reject the data.

Figure 3: (a) Correlation of the FWHMmeas. with the sum of Th, U, Ca and Pb (apfu) for 5 natural monazite samples. Points indicate individual measurement, and diamonds indicate the average value for each sample. The solid line in the figure



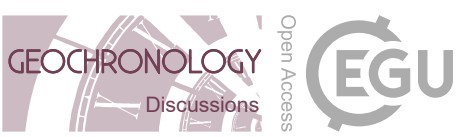

shows the calibration of the chemical broadening (FWHMcalc.) by Ruschel et al. (2012). The data shift along the vertical axis
195    as the structural disorder due to radiation damage. The structural disorder of monazite is partially recovered by self-irradiation,
so that structural disorder no longer progresses from a certain level (dashed line), and natural monazite does not undergo
metamictization (Nasdala et al., 2020). (b) Violin plots of combined with box plot of ΔFWHM.

### 4.4. Chemical etching of monazite fission-track

All five monazite samples underwent MFT etching procedures following the conditions outlined by Jones et al. (2019):
200    60−90 minutes in 6M HCl at 90°C. However, only MFTs of monazites in EY137-21 (Fig. 4e) achieved Dpar of 0.8 μm within
90 min. Monazites in MR3b and NR24-21 exhibited under-etched MFT, while Toya-5b and KRB19-B04 exhibited no
observable MFT. Subsequently, additional step-etching was employed for these four samples. Most Dpar reached 0.8 μm at
120 minutes for NR24-21(Fig. 4f), and 210 minutes for MR3b (Fig. 4d). For Toya-5b and KRB19-B04, several FTs appeared
approximately at 500 min. and the etching was finally completed at 1200 min. for Toya-5b (Fig. 4b) and 860 min. for KRB19-
B04 (Fig. 4c). It should be noted that for Toya-5b and KRB19-B04, the timing at which Dpar reached 0.8 μm for each FT
varies significantly from grain to grain, each with an error of about ±120 min. For both NR24-21 and EY137-21, intra-grain
heterogeneity in track density was observed. In these samples, the FT etching rate tended to be faster in the higher FT density
domains (Fig. 4e). Most grains in Toya-5b were zero-track and nine MFTs were observed in 61 grains.

### 5. Discussion

### 5.1. Inverse correlation between radiation damage and fission-track etching time on monazite

The degree of radiation damage ($\Delta_{FWHM}$) generally exhibits a positive correlation with sample age, except for EY137-21
(ca. 98 Ma) and NR24-21 (>105 Ma). Jones et al. (2023) reported cooling ages (zircon and apatite U-Th/He, and apatite and
monazite FT ages) for NR24-21 that is younger than those of EY137-21. Notably, the radiation damage in monazite can
undergo thermal annealing, similar to MFT (Seydoux et al., 2002), suggesting that $\Delta_{FWHM}$ may have a positive correlation not
with sample age but with the timing of cooling at a specific temperature. As mentioned above, the order between $t_{MFT}$ is Toya-
5b > KRG19-B04 > MR3b > NR24-21 > EY137-21, demonstrating an inverse correlation with $\Delta_{FWHM}$. As shown in Figure 4a,
these parameters have obvious inverse proportion.

This correlation mirrors the inverse relationship observed for zircon FT etching time and spontaneous FT density
(Krishnaswami et al., 1974; Gleadow et al., 1976). Previous studies have considered FT density as a proxy for U concentration,
positing that zircon with a higher degree of radiation damage exhibits a more rapid FT etching progression. The accelerated



etching rate along the FT, relative to the etching rate of the bulk crystal, primarily accounts for the FT etching time as the duration required for etching bulk crystals. The inverse correlation between FT etching time and spontaneous FT density is interpreted as reflecting the diminished chemical resistance of bulk zircon crystals resulting from radiation damage.

The inverse correlation between $t_{MFT}$ and $\Delta_{FWHM}$ revealed in this study can be explained by the same function as that observed in zircon FT. Similar to zircon, etching along MFT is assumed to proceed rapidly after immersion in HCl (5−15 min.: Jones et al., 2022). Consequently, the MFT etching time is predominantly considered as the duration necessary for bulk etching of the monazite crystal. It is reasonable to explain this phenomenon by suggesting that the etching time has been shortened in response to the diminished chemical resistance of bulk crystals, a consequence of monazite self-irradiation.

Meanwhile, the correlation between the chemical composition of monazite and $t_{MFT}$ is unclear from the results of this study
because the monazite samples have a similar chemical composition. Monazite, being a phosphate mineral, exhibits considerable variability in solid solution composition (Ce-La-Nd-Pr) and trace element composition. This variability may result in differences in etching rates due to composition, similar to apatite (e.g., Burtner et al., 1994, Ravenhurst et al., 2003). Despite the absence of significant differences in solid solution composition, particularly the average content of non-formula cations (Th, U, Ca, and Pb), between KRG19-B04, MR3b, and NR24-21, there were notable differences in $t_{MFT}$ among the
monazites in this study. In these samples, the variation in $t_{MFT}$ primarily signifies the influence of radiation damage rather than chemical composition. A similar investigation using monazite samples exhibiting a range in solid solution composition is required to clarify the specific impact of chemical composition on $t_{MFT}$. It remains challenging to ascertain whether the dominant factor is chemical composition or radiation damage. Nevertheless, the results of this study strongly indicate that radiation damage significantly influences MFT etching time.

**5.2. Etching of young and ordered monazite**

In the present study, MFTs in samples other than EY137-21 exhibited under-etching when subjected to the etching protocol outlined in the previous study (6M HCl, 90°C, 60–90 min.: Jones et al., 2019). This outcome can be ascribed to the comparatively lower radiation damage levels in the samples employed here compared to those utilized in previous studies. Monazites from previous studies, such as the Antsirabe monazite (ca. 500 Ma) and the Harcourt monazite (ca. 370 Ma), are
older than the monazites examined in this study and are presumed to exhibit more advanced radiation damage (Weise et al., 2009; Jones et al., 2019). While earlier investigations have verified the complete annealing of MFTs through pre-annealing, the conditions employed (at 400°C for 24 and 18 hours: Weise et al., 2009 and Jones et al., 2019, respectively) are inadequate for achieving total annealing of radiation damage (which requires 10 days at 900°C: Seydoux et al., 2002). Consequently, it is presumed that a certain degree of radiation damage persists in these samples. Moreover, the structural disorder of monazite is
partially recovered by self-irradiation, so that structural disorder no longer progresses from a certain level (ca. 12 cm$^{-1}$ in





$\Delta_{FWHM}$), and natural monazite does not undergo metamictization (Nasdala et al., 2020). As a result, the structural disorder in monazites older than a certain age tends to concentrate at a specific degree of radiation damage. For such samples, MFT etching is considered feasible within 60–90 min., as proposed in previous studies.

On the contrary, younger monazites with lower levels of radiation damage exhibit high chemical resistance, and existing
etching protocol fail to adequately etch MFT. Fitting the $t_{MFT}$ and $\Delta_{FWHM}$ data with an inverse proportional function using the least squares method yields a coefficient of 637.08 (Fig. 4a: $r^2$ score is 0.97). This equation indicates a rapid increase in $t_{MFT}$, particularly in low-damaged monazite with $\Delta_{FWHM}$ below 2 cm$^{-1}$. Even Miocene eruptive rocks required more than double the standard etching time (210 min. for MR3b), indicating that current etching protocols may prove insufficient for Cenozoic geological samples. Furthermore, radiation damage of monazite undergoes thermal annealing. This implies that even if the
crystallization age of monazite is sufficiently old, similar challenges in etching may arise if the sample experienced medium-to high-temperature conditions during the Cenozoic.

When employing the MFT method on rock samples with more recent formation and cooling ages in active tectonic settings, it is necessary to develop protocols that effectively etch chemical-resistant monazite. In the case of zircon, characterized by a similar diversity in etching times as observed in monazite, previous studies have defined the etching termination criteria and
implemented a step-etching method until the criteria are fulfilled (e.g., Yamada et al., 1993; 1995). As the application of MFT expands, it is recommended to investigate termination criteria specific to MFT etching and adopt a step-etching approach.

## 6. Conclusion

In this study, we conducted measurements of Raman spectra and chemical composition of monazite samples spanning from the Cretaceous to the Quaternary and compared the results with MFT etching experiment. The results reveal an inverse
proportion between MFT etching time and the degree of radiation damage in monazite, while the correlation between MFT etching time and chemical composition remains unclear. The results of the present study confirm previous considerations that the etching rate of MFT is predominantly influenced by radiation damage. On the other hand, young samples with lower levels of radiation damage exhibit higher chemical resistance, indicating that existing etching recipes may not sufficiently etch MFT. Future applications of MFT dating to younger geological samples will necessitate the determination of MFT etching
termination criteria and the adoption of step-etching procedures.

## Supplement

## Author contributions

Rock samples were collected by TN, SF, SS, SN. Mineral separation was conducted by TN and TD. Raman spectroscopic measurement and data analysis were conducted by TN and TK. Electron probe micro analysis was carried out by TN. MFT etching experiment was conducted by SF and TN. This study was conceptualized by TT and TD.

## Competing interests

At least one of the co-authors is a member of the editorial board of Geochronology

## Acknowledgements

We would like to thank Hideki Iwano (Kyoto Fission-Track Co. Ltd,) and Etienne Skrzypek (University of Graz) for providing an important information for monazite samples. The Raman spectroscopic measurement was carried out with the assistance of Fumiko Higashino (Kyoto University).

## Financial support

This study was funded by the Ministry of Economy, Trade and Industry, Japan as part of its R&D supporting program titled "Establishment of Technology for Comprehensive Evaluation of the Long-term Geosphere Stability on Geological Disposal Project of Radioactive Waste (JPJ007597) (Fiscal Years 2023)."

## Review statement

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





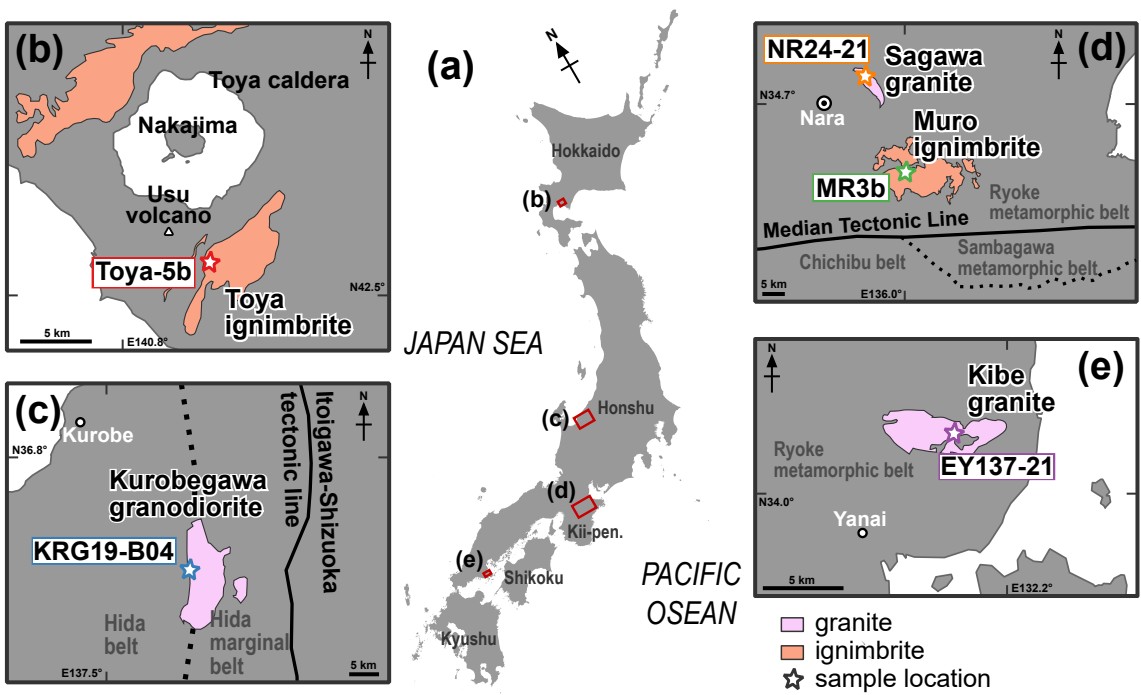

**Figure 1**: Geological units from which rock samples were collected in this study and a map of the sampling sites. (a) Index map. (b) The Toya ignimbrite is distributed around the Toya caldera in western part of Hokkaido. (c) The Kurobegawa granodiorite is exposed in the Hida mountain range in central Honshu Island. (d) The Sagawa granite and the Muro ignimbrite are distributed to the northern part of the Kii Peninsula, Honshu Island. (e) The Kibe granite is exposed in southwestern Honshu Island.



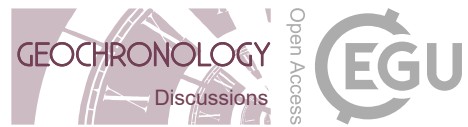

**Figure 2**: Photomicrographs and BSE images of representative monazite grains. (a, b) a colorless euhedral monazite in Toya-5b from the Toya ignimbrite. A Glassy melt inclusion and zircon often appear as an inclusion. BSE image show the homogeneity of the intra-grain composition. (c, d) a colorless and anhedral monazite in KRG19-B04 from the Kurobegawa granodiorite. Dashed line exhibits the boundary between primary and secondary domains observed in the BSE image. (e, f) a yellow-colored euhedral monazite in MR3b from the Muro ignimbrite. Monazite contains mineral inclusions of quartz andzircon, as well as polyphase solid inclusions composed of quartz, feldspar, and magnetite. BSE image shows the compositional homogeneity within the grain. (g, h) a yellow-colored, semi-euhedral monazite in EY137-21 from the Kibe granite. Internal structure of oscillatory zoning and/or sector zoning are observed under the BSE image. (i, j) a yellow-colored, semi-euhedral monazite in NR24-21 from the Sagawa granite. Internal structure of sector zoning are observed underthe BSE image. Mineral abbreviation follows that of Warr (2021).





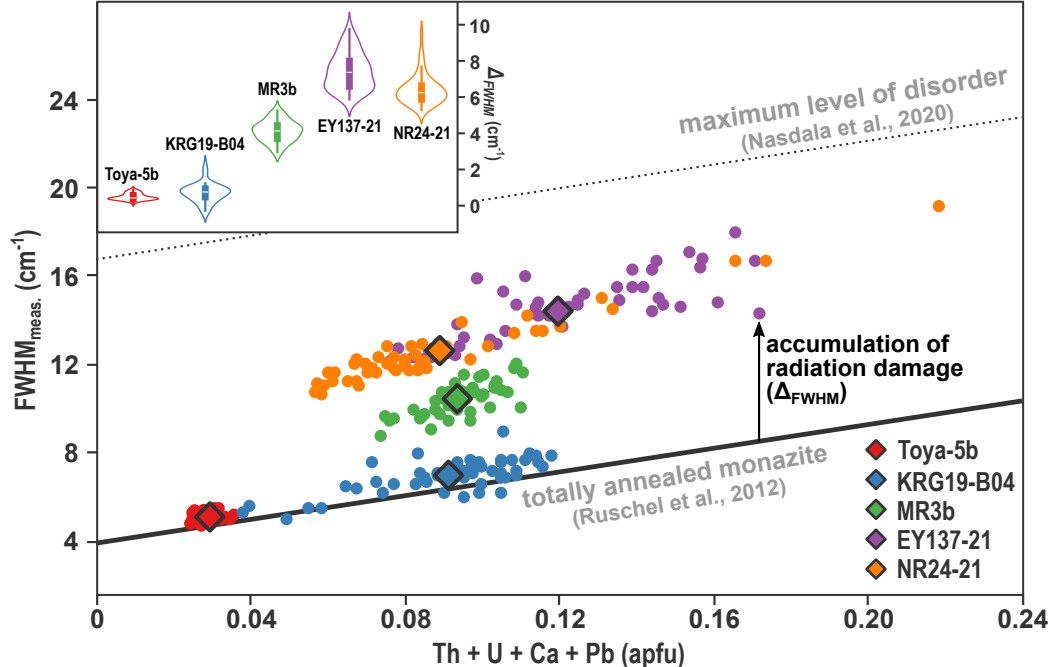

**Figure 3:** (a) Correlation of the FWHM$_{meas.}$ with the sum of Th, U, Ca, and Pb (apfu) for 5 natural monazite samples. Points indicate individual measurement, and diamonds indicate the average value for each sample. The solid line in the figure shows the calibration of the chemical broadening (FWHM$_{calc.}$) by Ruschel et al. (2012). The data shift along the vertical axis as the structural disorder due to radiation damage. The structural disorder of monazite is partially recovered by self-irradiation, so that structural disorder no longer progresses from a certain level (dashed line), and natural monazite does not undergo metamictization (Nasdala et al., 2020). (b) Violin plots of combined with box plot of $\Delta_{FWHM}$.



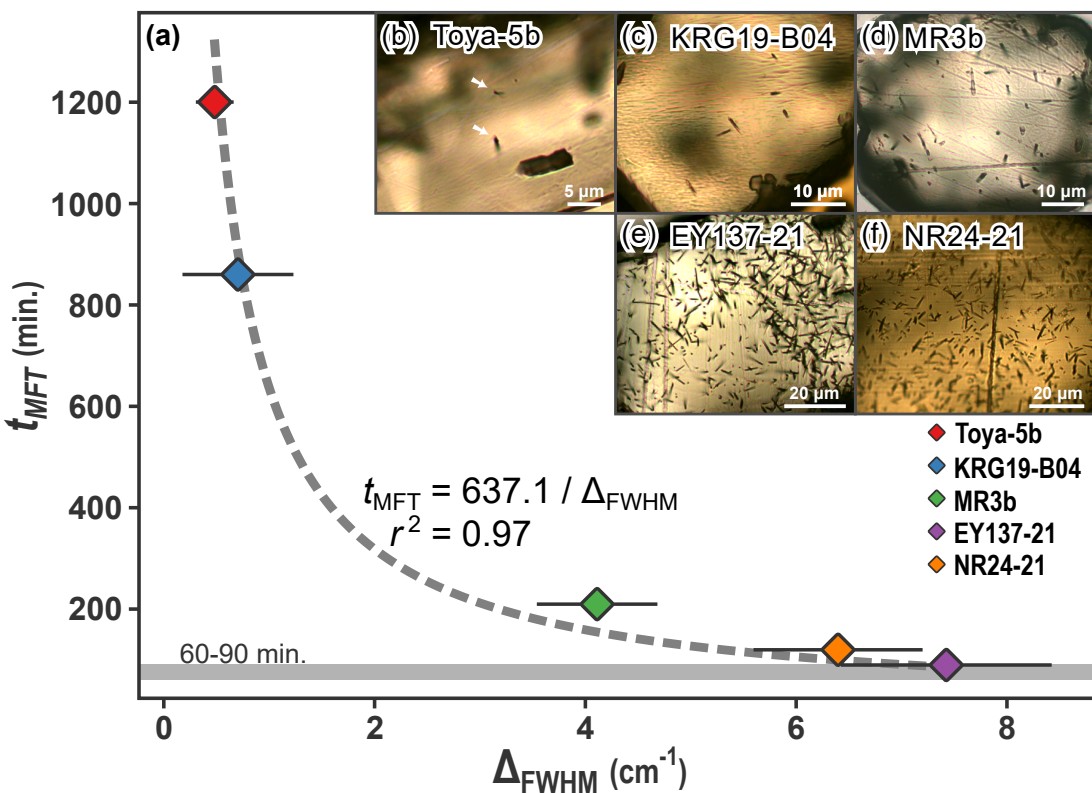

**Figure 4**: (a) Correlation of the $t_{MFT}$ with the $\Delta_{FWHM}$ for 5 natural monazite samples. The dashed line exhibits the fitted invers proportional equation using the least squares method, with the equation and $r^2$ score shown in the center of the figure. The gray band shown at the bottom of the diagram represents the etching time indicated in previous studies (Jones et al., 2019). (b–f) Photomicrographs of etched MFT in (b) Toya-5b, (c) KRG19-B04, (d) MR3b, (e) EY137-21, and (f) NR24-21.





**Table 1. Coordinates and lithology of rock samples used in this study.**

| sample name | N lat. (degree) | E lon. (degree) | unit name | rock type | reference age* |
|---|---|---|---|---|---|
| **Toya-5b** | 42.5276 | 140.8786 | *Toya Ignimbrite* Unit 5b (Goto et al., 2018) | pyroclastic flow deposit (monazites are mainly obtained from a white pumice) | 113.5 $^{+5.4}_{-5.2}$ka (Zrn and Mnz U-Th: Niki et al., 2022) ca. 108 ka (Zrn U-Th: Ito et al., 2014) |
| **KRG19-B04** | 36.6430 | 137.6860 | *Kurobegawa Granodiorite* lower unit (Harayama et al., 2000) | biotite-hornblende granodiorite | ca. 0.8 Ma (ZUPb: Ito et al., 2013; Suzuki et al., 2022) < 1 Ma (ZFT and ZHe: Yamada, 1999; King et al., 2023) |
| **MR3b** | 34.5565 | 136.0411 | *Muro Ignimbrite* | felsic ignimbrite | ca. 15 Ma (AFT: Iwano et al., 2009) 15.2 ± 0.5 Ma (ZFT: Iwano et al., 2007) |
| **EY137-21** | 34.0287 | 132.1632 | *Kibe Granite* | biotite granite | ca. 98 Ma (ZUPb: Skrzypek et al., 2016) 72.3–68.2 Ma (ZHe: Jones et al., 2023) 60.3 ± 1.5 Ma (AFT: Jones et al., 2023) 72.5–70.0 Ma (AHe: Jones et al., 2023) 2.2 ± 0.1 Ma (MFT: Jones et al., 2023) |
| **NR24-21** | 34.7224 | 135.9414 | *Sagawa Granite* | biotite granite | ca. 113–105 Ma (Bt Rb-Sr: Ishizaka, 1966) 75.8–37.3 Ma (ZHe: Jones et al., 2023) 53.4 ± 1.2 Ma (AFT: Jones et al., 2023) 83.4–41.0 Ma (AHe: Jones et al., 2023) 1.7 ± 0.1 Ma (MFT: Jones et al., 2023) |

*ages reported from the same location or stratigraphic unit are given with errors (95% confidence levels or 1 sigma), whereas ages reported from the same geologial unit are given as approximate age values. ZUPb: zircon U-Pb date; ZFT: zircon fission-track date; ZHe: zircon (U-Th)/He date; AFT: apatite fission-track date; AHe: apatite (U-Th-Sm)/He date; MFT: monazite fission-track date.





**Table 2. Representive results of Raman and electron probe micro analyses, calculated $\Delta_{FWHM}$, and $t_{MFT}$ for monazites in 5 rock samples.**

| sample name | spot | electron probe micro analysis (wt%) | | | | | | | | | | | | | | | | | | | | | Th+U+Ca+Pb (apfu) | Raman analysis (cm⁻¹) | | $\Delta_{FWHM}$ (cm⁻¹) | $t_{MFT}$ (min.) |
|---|---|---|---|---|---|---|---|---|---|---|---|---|---|---|---|---|---|---|---|---|---|---|---|---|---|---|---|---|
| | | $SiO_2$ | $P_2O_5$ | $CaO$ | $Y_2O_3$ | $ThO_2$ | $UO_2$ | $PbO$ | $La_2O_3$ | $Ce_2O_3$ | $Pr_2O_3$ | $Nd_2O_3$ | $Sm_2O_3$ | $Eu_2O_3$ | $Gd_2O_3$ | $Tb_2O_3$ | $Dy_2O_3$ | $Ho_2O_3$ | $Er_2O_3$ | $Tm_2O_3$ | $Yb_2O_3$ | Total | | Shift | $FWHM_{meas}$ | | |
| Toya-5b | spot 33 | 0.40 | 28.98 | 0.23 | 1.57 | 2.15 | 0.01 | n.d. | 11.31 | 30.38 | 2.95 | 14.26 | 2.14 | n.d. | n.d. | 0.53 | 0.77 | n.d. | n.d. | n.d. | n.d. | 99.90 | 0.029 | 972.61 | 5.32 | **0.59** | **1200** |
| KRG19-B04 | spot20 | 1.30 | 27.29 | 0.42 | 2.09 | 7.52 | 0.10 | n.d. | 11.67 | 30.02 | 2.65 | 9.98 | 1.41 | n.d. | 3.75 | 0.49 | 0.52 | 0.01 | n.d. | n.d. | n.d. | 99.19 | 0.088 | 973.41 | 7.05 | **0.76** | **860** |
| MR3b | spot 38 | 0.28 | 29.69 | 0.98 | 3.10 | 5.60 | 0.04 | n.d. | 9.60 | 28.47 | 2.64 | 11.34 | 2.13 | n.d. | 4.29 | 0.88 | 0.81 | n.d. | 0.01 | n.d. | n.d. | 99.86 | 0.091 | 972.76 | 10.81 | **4.42** | **210** |
| EY137-21 | spot06 | 1.44 | 27.73 | 0.58 | 1.90 | 8.29 | 0.17 | n.d. | 10.78 | 29.31 | 2.66 | 10.32 | 1.73 | n.d. | 3.88 | 0.55 | 0.56 | 0.02 | n.d. | n.d. | n.d. | 99.92 | 0.102 | 971.43 | 13.23 | **6.57** | **90** |
| NR24-21 | spot18 | 0.85 | 28.63 | 0.71 | 2.41 | 5.52 | 0.19 | n.d. | 11.80 | 30.20 | 2.59 | 9.76 | 1.50 | n.d. | 3.80 | 0.60 | 0.47 | 0.01 | n.d. | n.d. | n.d. | 99.04 | 0.088 | 971.55 | 12.52 | **6.39** | **120** |