# Peer review of "Short communication: Inverse correlation between radiation damage and fission-track etching time on monazite"

_Geochronology, 2024_

## Author Response (AR1)

**Reply to Comments by Sean Jones**

Thank you very much for providing important insight. In accordance with your comments, we have revised the original manuscript. The following is a response to the specific comments and technical corrections we received and the clarification of the revised text.

**Specific Comments**

1. In the abstract, please clarify that cooling age relates to unit age (e.g. crystallization age) rather FT cooling age.

■ As we discussed in the beginning of section 5.1, the corresponding ages to radiation damage are not the formation (crystallization) ages of the geological unit but the cooling and eruption ages indicated by ZHe, ZFT age. Accordingly, we have revised the wording of the abstract to clarify it.

2. The study of Jones et al. 2023 did use an initial starting length of 10.6 mm.

■ This information has been added in section 2.4.

3. Did the authors use a grinding plate in your sample preparation to expose an internal surface as well? If so, please include.

■ We used a grinding plate for the preparation and added a note at the end of section 3.1.1.

4. The authors note monazite of different colors (colorless – yellow – orange), as a side observation, did they find any relationship between chemical composition and/or radiation damage and color? Zircon is known to change color with increasing levels of radiation damage.

■ As you have pointed out, there seems to be some relationship between radiation damage and the color of monazite crystals. For example, monazites from Toya-5b and KRG19-04, which are less radiation damaged, are colorless, whereas monazite in MR3b is yellowish and monazites in Cretaceous granites (EY137-21 and NR24-21) show yellow- or orange-colored. As with etching time, the effect of chemical composition on color is unknown so far.

However, even synthetic monazite without radiation damage is known to show yellow color (e.g., Cherniak and Pyle, 2008). Therefore, the color of monazite may reflect some factor other than radiation damage. As this is currently unclear, we do not include this discussion in the revised manuscript.

5. In the methods section, the authors note that 50 Raman points were selected. Was this in total across for each sample or total for all samples? Please make clearer.

- We analyzed 50 points for each sample, and we have made this point clearer in the revised manuscript.

6. Dpar is good and universal terminology used to describe track openings for uniaxial minerals such as apatite. However, it is suggested that the authors adopt the terminology proposed in Jones et al. 2021 for describing track openings in monoclinic minerals (e.g. DPc, Dpb, Dpa).

- We agree that Dpa, Dpb, and Dpc are appropriate for monazite, rather than Dpar and Dper. Since the parameters we measured correspond to Dpc, we have revised the descriptions in the manuscript.

7. What was the reason for the low dispersion on the Raman Spectroscopic measurement in Toya-5b? It might be worth noting why this is the case.

- The low dispersion of the Raman data for monazite in Toya-5b can be explained by its homogeneous chemical composition. This is newly mentioned in section 4.3.

8. It would be nice to include in the results section the frequency in which the samples were checked during step-etching.

- We have added an additional note to section 4.4 on step etching intervals.

9. It is interesting that no relationship was found between chemical composition and etching time and continued research into this is likely required.

- Thank you for your informative comment. We believe that we should collect more monazite with various chemical compositions and conduct similar analyses to investigate the effect of chemical composition on etching behavior.

10. The effects of radiation damage on etching times are quite telling and it is obvious that step-etching is required working on unknown samples. It maybe that the original etchant used (12M HCl at 90°C for 45 min, Shukoljukov & Komarov, 1970) may be more suitable at etching younger, more resistant monazite due to being more concentrated and aggressive etchant (potentially reducing etching time).

- We agree that etching with 12M HCl is a useful approach for a young monazite. We chose to use 6M HCl in this study because of the possible degradation of the epoxy due to the concentrated acid. We would like to investigate this point and intend to report on it in a later paper.

11. Along with step-etching, it may be worth making multiple mounts when conducting application studies so that different etching domains are not over-etched. This would reduce bias in the results enabling the analyst to obtain data for different age populations.

- Thank you for your informative comment. For samples with lower levels of radiation damage, slight difference in the degree of the radiation damage of each grain are expected to result large

heterogenies in the degree of FT etching (Fig. 5). Therefore, as you mentioned, the method of reducing bias by using multiple mounts is presumably important, especially for FT dating.

**Technical Corrections**

Line 11: fission tracks in monazite

■ This error has been corrected in accordance with your comment.

Line 16: and the etching time

■ This error has been corrected in accordance with your comment.

Line 42: Make clear that you are talking about other minerals that have been well established in the FT technique (i.e. apatite, zircon). E.g. replace their with "the protocols of zircon/apatite….."

■ This error has been corrected in accordance with your comment.

Line 190: "did not reject data" is not clear. Please re-word.

■ The corresponding part of the manuscript has been corrected.

Figure 2 caption: Need space between the words "and zircon"

■ This error has been corrected in accordance with your comment.

Figure 3 caption: Maybe include "increases" (or similar) after the words structural disorder.

Reference

■ This error has been corrected in accordance with your comment.

Again, thank you for giving us the opportunity to improve our manuscript with your valuable comments. We hope that these revisions persuade you to accept our submission.

**Reference**

Cherniak, D.J., and Pyle, J.M.: Th diffusion in monazite, Chem. Geol., 256, 1-2, 52–61, https://doi.org/10.1016/j.chemgeo.2008.07.024, 2008.

**Reply to Comments by Birk Härtel**

We wish to express our appreciation to you for your insightful comments on our paper. In accordance with your comments, we have revised the original manuscript, figures, and tables. The following is a response to the general comments and line-by-line comments.

**General comments:**

1.  The justification of using the Dpar as stopping criterion for etching: I am not sure if assuming a constant Dpar as a stopping criterion is reasonable giving the differences in chemical composition and lattice disorder we observe in natural monazite.

■   Thank you for your important comment. As you pointed out, we did not adequately explain the justification for using Dpar (changed to Dpc following another reviewer's comment) as an index of MFT etching. Theoretically, etching needs to be performed until entire etchable length is etched and the measured FT density does not change. However, as it is difficult to determine this in practice, the width of the FT has long been adopted as the etching index for zircon over the past few decades. Jones et al. (2022) reported that the Dpc of MFT upon etch completion is approximately 0.8 μm. While the termination conditions for FT dating require more careful consideration, for the sake of convenience, this study adopted Dpc of 0.8 μm as the termination criterion. This is newly stated in section 3.2 of the revised manuscript.

2.  The authors do not mention on which material and band they carried out the instrument calibration.

■   We agree that the explanation in section 3.1.2 was inadequate. Also, as pointed out in a later line-by-line comment, the description contained a simple error. We have corrected the description, including the instrument calibration.

3.  There is no information on how they corrected their measured Raman bandwidths (FWHM) for instrumental broadening. This is especially important as the authors use the FWHM as a measure for radiation damage.

■   We did not perform the instrumental broadening correction in the original manuscript. As you pointed out, this correction is necessary, so we recalculated the ΔFWHM values by applying a correction based on the empirical formula of Váczi (2014) for each measurement. Accordingly, we corrected the diagram in Fig. 3a, the violin plot in Fig. 3b, and the diagram and regression curves in Fig. 5. This is a crucial point, and we would like to sincerely appreciate this comment.

**Line-by-line comments**

L. 30: contains

■ This error has been corrected in accordance with your comment.

L.42f.: Please consider reformulating this sentence. It would be good to clarify that the protocols refer to fission-track etching.

■ We have changed the sentence to clarify that this refers to zircon and apatite FT.

L. 87: BSE

■ This error has been corrected in accordance with your comment.

L. 112: was

■ This error has been corrected in accordance with your comment.

L. 120: Seydoux-Guillaume

■ This error has been corrected in accordance with your comment.

L. 128: comma missing after Meldrum et al.

■ This error has been corrected in accordance with your comment.

L. 142: The spectral resolution of 0.01 cm-1 seems very unrealistic to me. Given the laser wavelength and optical grid used I would expect a resolution > 1 cm-1. How was it determined or tested?

■ The reviewer's comment is correct. This is a simple error; $1$ $cm^{-1}$ is the correct value.

L.145: spectra

■ This error has been corrected in accordance with your comment.

L. 147: It would be good to include a description on how the spectra were corrected for the Raman baseline.

■ We have added the description of baseline correction in section 3.1.2.

L. 154: Please include information on the instrumental-drift correction of the EPMA data.

■ We did not perform an instrumental drift correction as we obtained our standards in the same session as the analysis of the unknown sample.

L. 161: Is there a relationship between grain chemistry or radiation damage and Dpar? If future studies also use Dpar as a criterion for full etching of monazite, it would be good to have an indication of its variation.

■ Since Dpar (Dpc) was not measured at the same etching time between each sample, a comparison cannot be obtained. Meanwhile, since Dpar is an index of etching progress, it is expected that the sample with higher levels of radiation damage will have a larger Dpar value when compared at the same etching time (positive correlation). Since such a comparison was not actually made, this description was not included in the revised manuscript. However, we would like to make such a comparison in further studies.

L. 191: Could these data with negative ΔFWHM also be explained by measurement uncertainty instead of only the uncertainty of the chemical-broadening trend?

- As you pointed out, both measurement errors and the propagation of calibration errors may have contributed to the negative value. We have modified our original sentence to agree with this point.

L. 192f.: It looks like a figure caption has been inserted here.

- We have deleted this sentence.

L. 230: As I see it, the Toga sample has a significantly and less variable chemical composition that the other samples.

- This statement is a general remark and is not intended for a specific sample. The sentence has been corrected to clarify it.

L. 256: Please cite the equation for the inverse relationship here.

- This error has been corrected in accordance with your comment.

L. 257: If I got it right, there should be a decrease of tMFT with increasing $\Delta$FWHM, not an increase.

- This error has been corrected in accordance with your comment.

Figure 4: The micro-photographs of the etched monazite surfaces are quite small. The authors might consider putting them into a separate figure.

- In accordance with your comment, the photographs have been divided into Fig. 4 and the diagram into Fig. 5. The description of Fig. 4 in the manuscript has been partially revised accordingly.

Figure 4 caption, line 1: inverse

- This error has been corrected in accordance with your comment.

Table S6 and following: Please specify that the the table refers to the $\nu1(PO_4)$ Raman band.

- In accordance with your comment, we have changed this in Table 2 and Table S6.

Again, thank you for giving us the opportunity to improve our manuscript with your valuable comments. We hope that these revisions persuade you to accept our submission.

**Reference**

Hasebe, N., Tagami, T., and Nishimura, S.: Towards zircon fission-track thermochronology: Reference framework for confined track length measurements, Chem. Geol., 112, 169–178, https://doi.org/10.1016/0009-2541(94)90112-0, 1994.

Jones, S., Kohn, B., and Gleadow, A.: Etching of fission tracks in monazite: Further evidence from optical and focused ion beam scanning electron microscopy, Am. Mineral., 107, 1065–1073, https://doi.org/10.2138/am-2022-8002, 2022.

Váczi T.: A new, simple approximation for the deconvolution of instrumental broadening in spectroscopic band profiles. Appl. Spectrosc., 68, 1274–1278, https://doi.org/10.1366/13-07275, 2014.

Yamada, R., Tagami, T., and Nishimura, S.: Confined fission-track length measurement of zircon: assessment of factors affecting the paleotemperature estimate, Chem. Geol., 119, 293–306, https://doi.org/https://doi.org/10.1016/0009-2541(94)00108-K, 1995.